# Convective dynamics in mantle of tidally-locked exoplanets

Daisuke Noto [1] ✉, Takehiro Miyagoshi [2] ✉, Tomomi Terada [3], Takatoshi Yanagisawa [3,4] & Yuji Tasaka [3,4] ✉

Tidal locking imposes distinctive thermal forcing on super-Earth exoplanets in habitable zones, i.e., permanent stellar flux forces extraordinary day-night temperature contrast. However, it may be premature to conclude that life is absent in such supposedly harsh environments—flaming hot on dayside and freezing cold on nightside—when accounting for unobservable features, such as internal convective dynamics and their consequential impact on the surface environment. We establish a simplistic but canonical framework scalable for modeling the convective dynamics in the mantle of tidally-locked exoplanets. The laboratory experiments unveiled an everlasting system-scale circulation that localizes mass and heat transport inside the mantle for a wide range of parameters. We identified the governing parameters that characterize the mass and heat transport of the system and demonstrated their significance. The permanently anchored internal convective structures will be integrated as extraordinary tectonic and deep core activities that differ substantially from those on Earth. In particular, a gradually varying heat flux distribution from the substellar to antistellar points hints at the presence of liquid water in the mid- to high-latitudes due to their moderate geothermal heating, which can potentially host and nurture life on such faraway worlds.

Most known terrestrial planets orbit near their parent stars with ultra-short periods, prone to being tidally locked[1]. A simple yet profoundly intriguing question is: 'Can such faraway worlds nurture and sustain life?' Although recent advances in telescopic observations have unearthed diverse surface conditions of individual super-Earth exoplanets, e.g., surface temperature and atmospheric conditions, they are largely uncertain, stirring active debates. For instance, the super-Earth 55 Cancri e (1.9 Earth radii) exhibits characteristics indicative of atmospheric circulation[2], but its interpretation has been revisited recently[3]. The smaller and cooler super-Earth LHS 3844b (1.3 Earth radii) has recently been identified as having an absence of a thick atmosphere[4], justifying theoretical predictions for the rare retention of substantial atmospheres on hot terrestrial planets orbiting small stars. These recent works underscore the considerable diversity of surface conditions on super-Earths, depending on various factors such as their sizes, compositions, and distances from parent stars[5–7]. Given that surface conditions—despite being observable—are marked by uncertainty and diversity, the nature of internal dynamics is even a deeper mystery. In particular, the mantle dynamics, primarily driving the surface tectonic activities, intertwines with the other factors and determines the consequential surface environments[8], yet is largely unexplored. Deepening the comprehension of internal dynamics in such distinctive environments—rare in our solar system—is, therefore, of great importance in searching for the potential existence of life, motivating model-based research based on a profound physical understanding.

Telescopic observations help us constrain the problem from the perspective of thermal forcing conditions. Tidal lock supports a

[1]Department of Earth and Environmental Science, University of Pennsylvania, Philadelphia, USA. [2]Research Institute for Value-Added-Information Generation, Japan Agency for Marine-Earth Science and Technology, Yokohama, Japan. [3]Laboratory for Flow Control, Faculty of Engineering, Hokkaido University, Sapporo, Japan. [4]Research Institute for Marine Geodynamics, Japan Agency for Marine-Earth Science and Technology, Yokosuka, Japan. ✉e-mail: dnoto@sas.upenn.edu; miyagoshi@jamstec.go.jp; tasaka@eng.hokudai.ac.jp

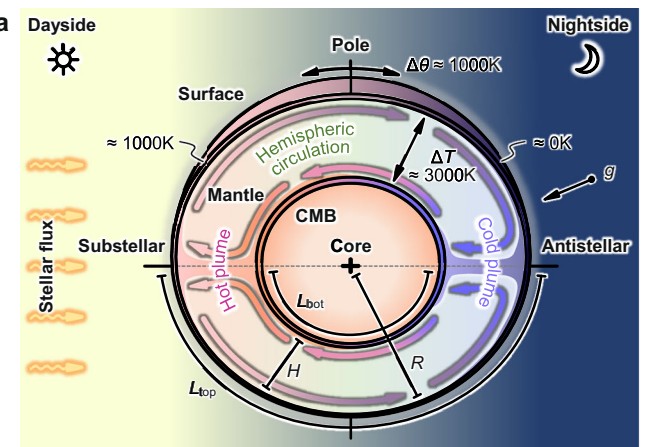

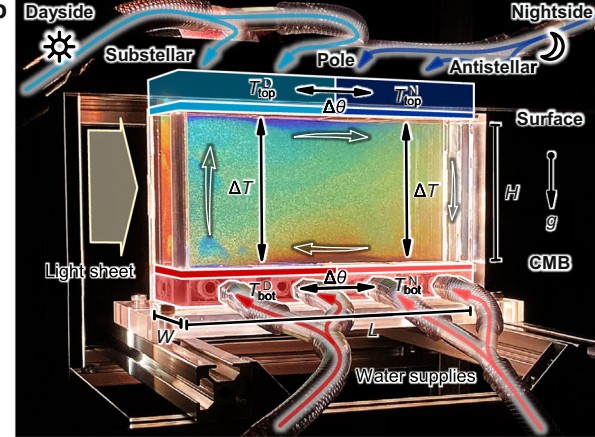

**Fig. 1 | Mantle dynamics of tidally-locked super-Earth exoplanets. a** Schematic of convective flows in a mantle layer of a tidally locked super-Earth exoplanet. Dayside faces its star permanently, leading to the surface temperature of $\approx 1000\,\text{K}$, whereas that of the nightside is absolute zero. **b** Laboratory realization of the tidally-locked hemispheric mantle to investigate its convective dynamics through optical visualization. The light sheet illuminates the TLC particles, exhibiting reddish for low and bluish for high temperatures, seeded in the glycerol solution. Four surface temperatures are controlled separately by supplying refrigerants from four independent thermostatic baths, achieving the stable imposition of horizontal and vertical temperature differences, $\Delta\theta$ and $\Delta T$.

tremendous temperature difference along a planetary surface, for instance, $\approx 1000\,\text{K}$ between the substellar and antistellar points on LHS 3844b[4], making the dayside hot and the nightside cold. Such unique thermal conditions will promote heterogeneous internal structures such as magma ponds and magma oceans localizing at the substellar regions[9,10], differing from those on Earth[8,11]. Some earlier numerical works are motivated by such uncommon thermal forcing and have implemented stellar flux for simulations of mantle convection[12–15]. These attempts successfully showcase potential convective scenarios on specific tidally-locked exoplanets, yet we still lack a unified understanding of the convective dynamics, i.e., 'What parameters fundamentally control the mantle convection?' To seek an answer to this question requires further generalization of the problem through establishing a dimensionless framework. Its significance is also relevant for other systems experiencing complex thermal forcing that arise across diverse environments in both natural systems[16–19] and industrial systems[20–22].

Here, we build a minimal but paradigmatic experimental system, specifically designed to mimic mantles of tidally-locked super-Earth exoplanets to deepen the comprehension of their convective dynamics on such distant worlds at fundamental levels. Carefully designed laboratory analog leads to the identification of different convective regimes through optical measurements and justifies theoretically derived governing parameters. Our dimensionless framework is scalable to mantle dynamics of tidally-locked exoplanets with various thermal conditions from laboratory experiments, covering their possible scenarios, as well as overarches the extensive knowledge of thermal convection and faraway worlds' dynamics. Our findings imply the unexpected potential of super-Earth exoplanets for supporting moderate thermal environments suited for life from a thermodynamic point of view, thanks to tidal locking, paving the future pathway for further investigations on such systems from multidisciplinary aspects.

## Results
### Formulation of problem

A paramount characteristic of a tidally locked exoplanet is its permanent exposure to stellar flux from its star on the dayside, imposing a considerable temperature gradient along the planetary surface in addition to the internal vertical gradient across the mantle. A potential mantle dynamics expected in such a thermodynamically unique system is schematized in Fig. 1a. The temperature difference $\Delta T$ between the colder surface and hotter core-mantle boundary (CMB) is imposed

parallel to the gravity across the mantle with a thickness of $H$, leading to an unstable vertical temperature gradient. Concurrently, the horizontal one can be represented by a circumferential length $L_{\text{top}}$ (or $L_{\text{bot}}$) and a day-night temperature difference $\Delta\theta$. Note that $\Delta T$ and $\Delta\theta$ are defined as effective temperature differences as a result of intertwined effects of surface and core dynamics. By doing so, physical interpretations derived from this framework become scalable to diverse systems irrespective of their stellar flux, surface, and core conditions. The system is expected to exhibit hemispherical symmetry and is considered to be a hybrid form of the paradigmatic problems of Rayleigh–Bénard convection (RBC)[23] and horizontal convection (HC)[24], offering us a firm background knowledge of process-understanding of the tidally-locked mantle dynamics.

What parameters govern such a hybrid system? We consider a Boussinesq fluid with the linear equation of state for simplicity, $\rho(T) = \rho_0\left[1 - \alpha(T - T_0)\right]$, where $\alpha$ is the thermal expansion coefficient and $\rho_0$ is the fluid density at a reference temperature $T_0$. All density anomalies produced at the surfaces, $\Delta\rho_z\,(=\alpha\rho_0\Delta T)$ and $\Delta\rho_x\,(=\alpha\rho_0\Delta\theta)$, exert as available potential energy, destabilizing the system[25,26]. Therefore, $H$ is naturally chosen as the characteristic length scale, as both RBC and HC drive full-depth circulations[23,24]. By considering conversion of potential energy to kinetic energy, $(\Delta\rho_z + \Delta\rho_x)gH \sim \rho_0 U^2$, the velocity scale $U$ is obtained as $U \sim \sqrt{\alpha(\Delta T + \Delta\theta)gH} = \sqrt{\alpha(1+\Theta)\Delta TgH}$, where $\Theta = \Delta\theta/\Delta T$ is the degree of thermal nonuniformity. Notice that $U$ can be written as $U = \sqrt{1+\Theta}\,U_z$, where $U_z = \sqrt{\alpha\Delta TgH}$ is the free-fall velocity defined in RBC. Accordingly, the hybrid Rayleigh number integrating $\Delta T$ and $\Delta\theta$ is defined as

$$\text{Ra} = \frac{g\alpha(\Delta T + \Delta\theta)H^3}{\kappa\nu} = (1+\Theta)\,\text{Ra}_z, \qquad (1)$$

where $\text{Ra}_z = g\alpha\Delta TH^3/(\kappa\nu)$ is the classic Rayleigh number for RBC. This definition of Ra implies a linear superposition of the extra buoyancy gained by $\Delta\theta$ on that by $\Delta T$. Assuming that the material properties and the internal structures, such as the total radius to the mantle thickness, are similar to those of the Earth, $\text{Ra}_z$ in the mantle of such a super-Earth —up to 10 Earth mass ($\approx 2$ Earth radii)—is estimated to be $\text{Ra}_z = \mathcal{O}(10^4 - 10^9)$ with $\Delta T = 3000\,\text{K}$. On the other hand, the observation revealed that the surface temperature at the dayside can reach $\approx 1000\,\text{K}$, whereas the nightside is $\approx 0\,\text{K}$[4,27], therefore, $\Delta\theta = 1000\,\text{K}$. These values $\Delta T$ and $\Delta\theta$ should vary significantly across planetary systems—$\Delta T$ may

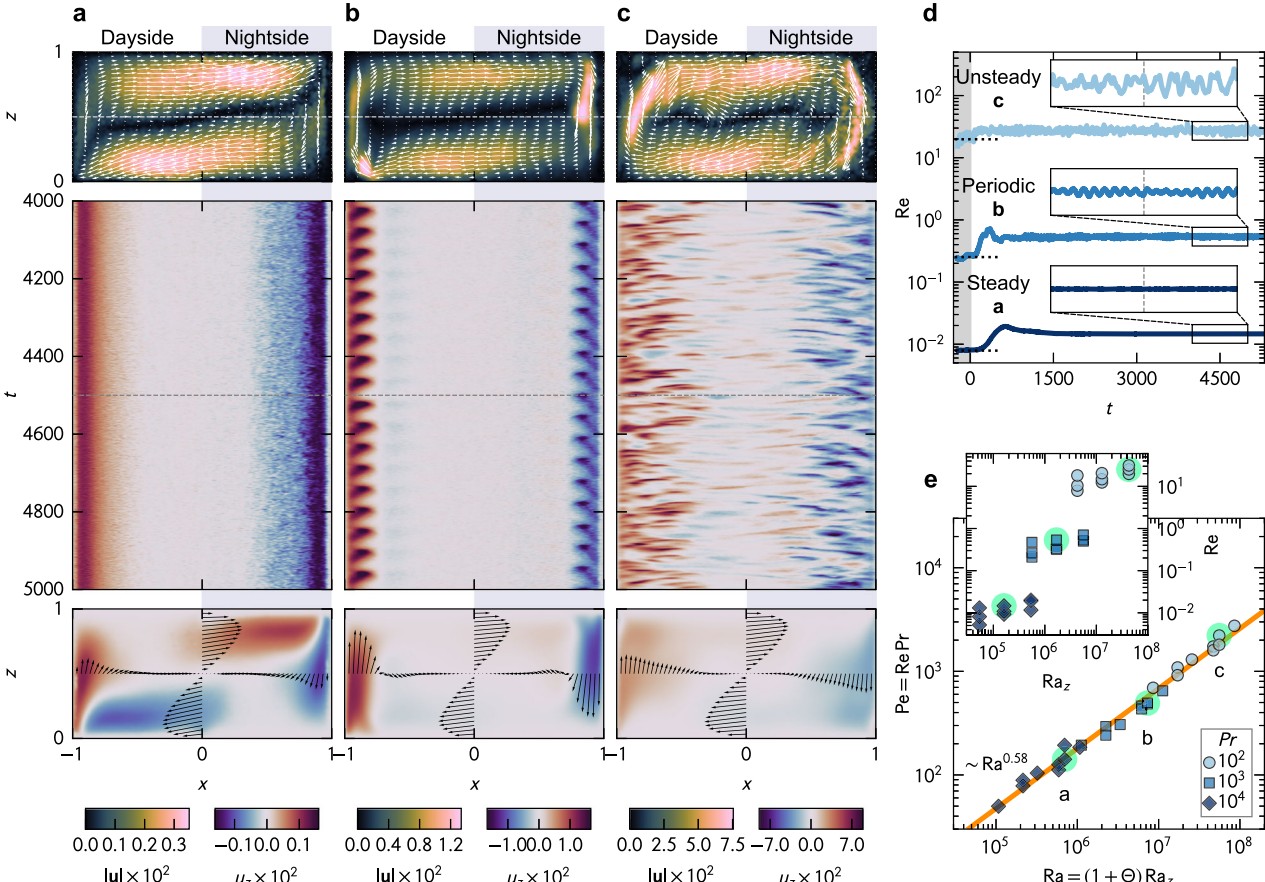

**Fig. 2 | Convective dynamics in the mantle of tidally-locked exoplanets obtained through the laboratory analog.** Snapshots of velocity fields (top), Hovmöller diagram of vertical velocity $u_z$ obtained at the middle depth $z = 0.5$ (middle), and time-averaged vertical velocity $u_z$ (bottom) for (**a**) steady, (**b**) periodic, and (**c**) unsteady conditions. The left half and the right half of each panel represent the dayside and the nightside, respectively. **d** Time evolution of the Reynolds number, Re, for the three conditions shown in (**a**, **b**, and **c**). Dotted horizontal lines represent Re obtained before imposing Θ, i.e., Rayleigh–Bénard convection. Each inset corresponds to the period shown in the Hovmöller diagram,

$4000 < t < 5000$. **e** Péclet number Pe = Re Pr plotted over the hybrid Rayleigh number Ra = (1 + Θ) $Ra_z$, collapsing the data. Colors correspond to Pr, and the conditions highlighted correspond to (**a**, **b**, and **c**). The solid line is the power-law curve, Pe ∝ $Ra^{0.58}$, acquired with the least-squares fitting. The inset shows Re plotted over the classical Rayleigh number, $Ra_z$, showing deviations originating from Θ and differences across Pr. All variables are nondimensionalized with the length scale $H$, the velocity scale $U$, and the time scale $H/U$. The original videos visualized using the TLC particles corresponding to (**a**, **b**, and **c**) are provided as *Supplementary Movies* 1, 2, and 3, along with the hemispherical projection.

approach 10000 K at most for massive super-Earths[28,29] and Δθ can range widely from 0 K (like Earth) to 2000 K (55 Cancri e)[3]—and depend also on the thermal evolutionary stage of the planet of interest. Another essential parameter is the Prandtl number Pr = ν/κ, representing the ratio of viscous and thermal dissipation, which is usually accounted to be infinity and therefore invariant owing to the extremely high viscosity of the rocky mantle, $\mathcal{O}(10^{20}\ \text{Pa} \cdot \text{s})$[30]. Although it is out of the present scope, the length-to-height aspect ratio $\mathcal{A} = L/H$ varies with planetary radius in the range $\mathcal{O}(1)$. The detailed discussion on the parameter ranges for tidally-locked super-Earths is provided in *Supplementary Information*, Supplementary Fig. S1.

Laboratory analog was a projection of the hemispheric mantle realized in a tabletop rectangular tank with transparent adiabatic sidewalls as showcased in Fig. 1b. The domain was thus $L = L_{top} = L_{bot}$ in length, and the gravity pointed downward. The left half ($x < 0$) and the right half ($x > 0$) corresponded to the dayside and nightside, respectively. The top (surface) and bottom (CMB) temperatures were controlled individually, and denoted as $T^D_{top}$ and $T^D_{bot}$ for the dayside (with a superscript D) and $T^N_{top}$ and $T^N_{bot}$ for the nightside (with a superscript N), and thus $T^D_{bot} - T^D_{top} = T^N_{bot} - T^N_{top} = \Delta T$ and $T^D_{top} - T^N_{top} = T^D_{bot} - T^N_{bot} = \Delta\theta$. We utilized aqueous glycerol solutions with variant concentrations as test fluids to achieve high Pr conditions, $Pr = \mathcal{O}(10^2 - 10^4)$, while maintaining their transparency. The

latter high Pr range can be essentially regarded as infinite in vigorous convection[31], modeling the rocky mantle's feature adequately. Flow fields were characterized at the middle vertical cross-section through image processing on micro-particles encapsulating thermochromic liquid crystals (TLC) seeded into the fluids[32–35]. The laboratory setup and the data analyses are elaborated in *Methods*. This setup covers a wide range of parameters, $Ra_z = \mathcal{O}(10^5 - 10^8)$ and $\Theta \in [0.1, 10]$, largely overlapping those for mantle on tidally-locked exoplanets estimated using characteristic thermophysical properties of Earth's mantle[30,36,37] (see *Supplementary Information*, Supplementary Fig. S1). The present framework, therefore, will delineate typical dynamics of such convective systems while maintaining generality.

### Convective dynamics of tidally-locked mantle

By running the experiments from the initial states—fully developed RBC without Θ—until the statistically steady states (SSSs) after imposing Θ, the laboratory analog successfully depicts convective dynamics as showcased in Fig. 2. Three characteristic convective regimes characterized by time dependency, steady (laminar), periodic (laminar-to-turbulent transition), and unsteady (turbulent), are shown in Fig. 2a, b, and c, respectively. Note that the original videos visualized using the TLC particles and their back-projection into a hemispheric system are provided as Supplementary Movies to graphically capture

their dynamics. The snapshots of the velocity fields shown at the top of each condition demonstrate the achievement of a system-scale clockwise circulation regardless of the convective regimes, akin to hemispheric circulations predicted elsewhere[9,10,12,14,15]. The time dependency, emerging at $Ra \gtrsim 4.7 \times 10^6$ (see *Supplementary Information*, Supplementary Fig. S2), can be evaluated from the Hovmöller diagram of vertical velocity $u_z$ extracted at $z = 0.5$ as illustrated in the middle panel. The upward and the downward flows concentrate at the left and the right walls, corresponding to the substellar and the antistellar points. Contrarily, the center of the domain, the high latitude region, scarcely holds such vertical flows, and horizontal exchange flows dominate. Such a localization effect of vertical flows remains identical even when the system shifts to the turbulent regime with higher Ra. Thermal plumes detached from the horizontal boundaries immediately get trapped in the overturning circulation and swept laterally. Time-averaged vertical velocity distributions displayed at the bottom distinctly depict the common mean structure, an overturning circulation, hidden by the localized small-scale ascending/descending plumes.

Remarkably, the system-scale circulations showcased in Fig. 2a, b, and c are observed for all the conditions explored in the experimental set. The latter suggests that such circulation is a characteristic convective pattern in tidally locked mantle, covering wide ranges of the parameters Ra and Θ. But, what is the mechanism of such a unique structure? In turbulent RBC for $Ra_z > 10^6$, a system-scale circulation, the so-called large-scale circulation (LSC), is renowned to form because of strong inertia arising from thermal plume emissions[23]. The circulation, however, manifests even in the laminar regime with small $Ra_z$, where no thermal plume is generated as shown in Fig. 2a, and so is LSC. Therefore, the circulation originates from baroclinic torque imposed on the system, the main driver of HC characterized with Θ, rather than the self-organized turbulent structures in RBC. In fact, for the turbulent regime with high $Ra_z$, the LSC initially formed before the imposition of Θ is wiped out by the clockwise overturning circulation of HC, irrespective of the initial LSC's direction. As a consequence, small-scale structures like plumes are transported laterally to the walls downstream, as shown in Fig. 2b and c, leading to the localization of substellar upward plumes and antistellar downward plumes. In turn, upstream regions, the top of the dayside and the bottom of the nightside, are stabilized. The experimental results, corroborated by the extensive knowledge, unravel a synergy effect of HC at the large scale and RBC at the small scale, determining the dynamics of mantle convection in tidally locked exoplanets.

## How vigorous is the convective motion?
Now, let's direct our attention to the global kinetic energy that characterizes the vigorousness of the convective activities in the mantle, ultimately influencing the surface tectonic and the deeper core activities. Since the system is reasonable to regard as quasi-two-dimensional for the strong overturning circulation, i.e., $|u_y| \ll |u_x|, |u_z|$, we can evaluate the mean specific kinetic energy of the system as $E_k = \langle u_x^2 + u_z^2 \rangle_V / 2$, where the operator $\langle \cdot \rangle_V$ denotes a volume average. The latter further allows defining the Reynolds number as $Re = U_{rms} H / \nu$, the ratio of consequential inertia and viscous forces, with the root-mean-squared velocity $U_{rms} = \sqrt{2E_k}$. The time evolution of Re for the three cases showcased in Fig. 2a, b, and c are plotted in Fig. 2d. The gray shade represents the period before the imposition of Θ, and the horizontal dotted lines indicate the corresponding Re values at the initial conditions (only RBC) for reference. Insets correspond to the period shown in the Hovmöller diagrams, $4000 < t < 5000$. The time dependency of each regime shows distinct differences in Re profiles, larger amplitude, and higher frequency for the turbulent regime (see the corresponding power spectra in *Supplementary Information*, Supplementary Fig. S2). However, all conditions achieve plateaus at higher Re after the initial transitions of

$t < 1000$, meaning that the addition of Θ increases the total kinetic energy of the system.

The system appears to possess higher Re values for higher Ra in general, but intriguingly, these values at the SSSs exhibit differences with orders of magnitude. Such enormous differences may be associated with those of Pr, as viscous dissipation becomes stronger as Pr increases, resulting in lower Re. Accordingly, the Péclet number $Pe = U_{rms} H / \kappa = Re \, Pr$, the ratio of advective and diffusive transports, offers an impartial comparison across different Pr conditions in evaluating the kinetic energy production rather than Re. We map the Pe values achieved at the SSSs versus Ra in Fig. 2e. The inset shows Re versus $Ra_z$ for reference. It is evident that $Ra_z$ solely does not grasp the increment of Pe (or Re) from the scattered symbols in the inset. Conversely, Pe exhibits an excellent collapse when plotted against Ra (see the comparison across different parameters in *Supplementary Information*, Supplementary Fig. S3). This remarkable collapse upholds the propriety of the proposed hybrid Rayleigh number Ra as the fundamental parameter. It also provides the best power-law fitting of $Pe \propto Ra^{0.58 \pm 0.23}$ as indicated by the orange solid line. Despite the different thermal forcing conditions, the obtained exponent, 0.58, agrees well with the existing scaling $Re \sim Ra_z^{2/3} Pr^{-1}$ established for the boundary-layer dominated large-Pr RBC, where the thickness of the viscous boundary layer is comparable with the system height $H$[38,39]. Although the latter scaling was originally designed for RBC, considering the kinetic energy production from the present system's available potential energy, $\sim (1 + \Theta) Ra_z$, the scaling is replaced with Ra spontaneously, leading to $Re \, Pr = Pe \sim Ra^{2/3}$ with the exponent similar to that in the experiments.

The collapse with Ra substantiates the essential role of the horizontal temperature gradient, as important as the vertical one, regarding mantle dynamics on tidally locked exoplanets. It implies that the vigorous convective activities, redistributing and mixing passive scalars like heat and chemicals within the mantle, can be sustained even with a small vertical temperature gradient. Planets usually diminish their internal vertical temperature gradients in the course of thermal evolution and are considered to weaken or eventually cease their convective and tectonic activities, like Mars[40–42]. However, tidally-locked exoplanets may maintain vigorous convection as they keep large Θ due to the perpetual exposure to the stellar flux. Moreover, although the presence of volcanic/tectonic activities in super-Earth exoplanets are still debated under the absence of horizontal temperature gradients because of various rheological modeling approaches[43,44], the significant role of Ra (or Θ) hints at unexpectedly vigorous volcanic/tectonic activities in super-Earth exoplanets.

## How much heat is transported?
Next, we shift our interest to heat distribution, an essential factor in considering the presence of liquid water, and ultimately, the habitability of planets. The question can be reformulated, whether geothermal heat allows the presence of liquid water. We unearthed that the thermal nonuniformity Θ plays a pivotal role in distributing heat in distinct ways. Mean temperature distributions are reconstructed from the mean velocity fields assimilated with the heat equation using the method proposed and validated earlier[34,35,45,46] (see *Methods* for the details). Examples for $Ra_z = 1.7 \times 10^6$ and $Pr = 10^3$ are showcased in Fig. 3a, b, and c, corresponding to Θ = 0.33, 1.00, and 3.33, respectively. The velocity vectors, the arrows superposed on the temperature field represented as colors, illustrate the overturning circulations. In the temperature distributions, the localization of upward and downward flows is reflected as that of hot and cold fluids at the left (substellar point) and the right walls (antistellar point). Such concentrations of fluids with strong temperature anomalies at the boundaries are less distinctive for Θ = 0.33. As Θ increases, the heat is more concentrated at the walls, and simultaneously, the downstream surfaces are exposed

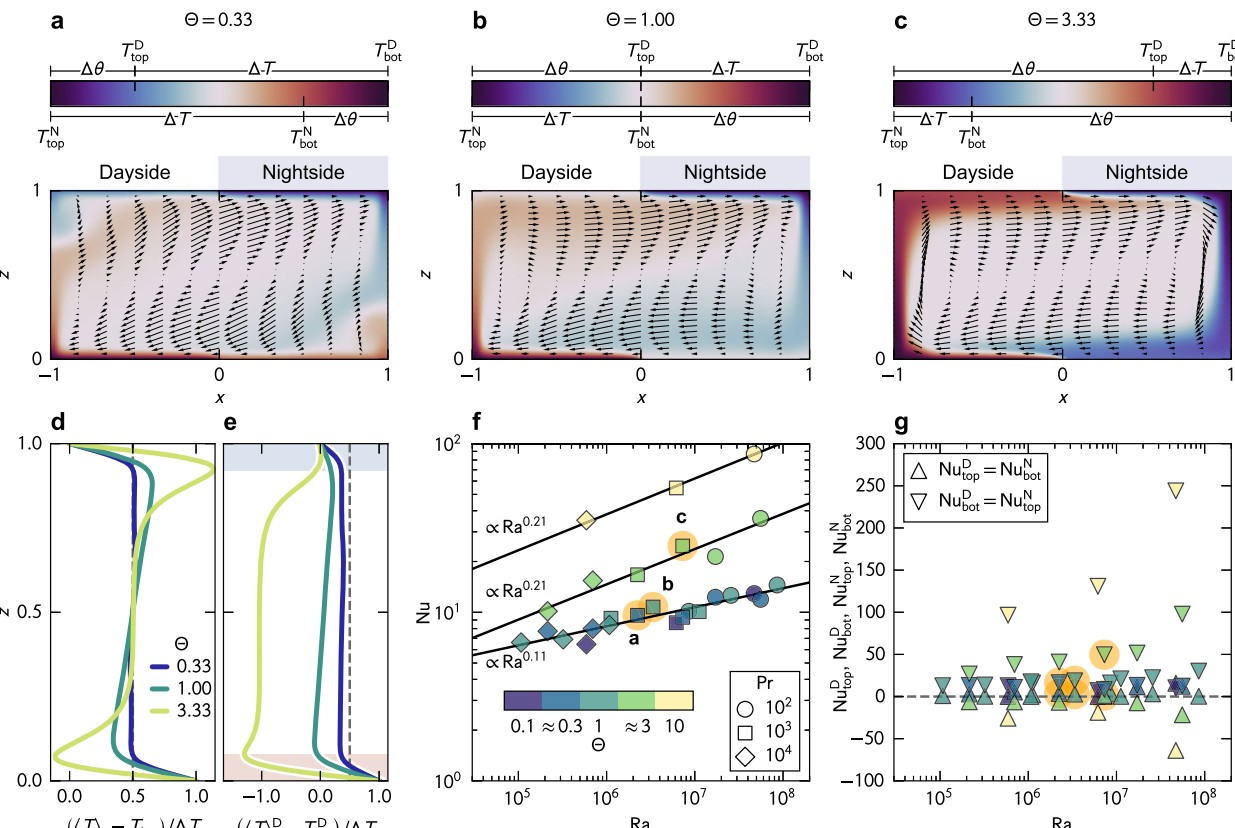

**Fig. 3 | Thermal structures of tidally-locked mantle.** Mean temperature distributions for (**a**) $\Theta = 0.33$, (**b**) $\Theta = 1.00$, and (**c**) $\Theta = 3.33$. Colorbars intricate the boundary conditions for each condition. **d** Horizontally averaged temperature profiles using the whole domain, corresponding to (**a**, **b**, and **c**). **e** Horizontally averaged temperature profiles using the dayside only. **f** The Nusselt numbers Nu versus the hybrid Rayleigh numbers Ra. Colors correspond to $\Theta$ as indicated by the colormap, and symbols represent Pr. Conditions highlighted with orange circles are

those shown as (**a**, **b**, and **c**). Solid lines are the best power-law fittings for the data with $\Theta \leq 1$ (Nu $\propto$ Ra$^{0.11}$), $\Theta \approx 3$ (Nu $\propto$ Ra$^{0.21}$), and $\Theta = 10$ (Nu $\propto$ Ra$^{0.21}$), obtained with the least-squares fitting. **g** The local Nusselt numbers $Nu_{top}^{D}$ and $Nu_{bot}^{D}$ (or $Nu_{bot}^{N}$ and $Nu_{top}^{N}$) versus the hybrid Rayleigh numbers Ra. Upward triangles are $Nu_{top}^{D}(=Nu_{bot}^{N})$ and downward triangles are $Nu_{bot}^{D}(=Nu_{top}^{N})$. The latter is a positive definite, whereas the former can be negative to compensate for the massive horizontal heat transport at large $\Theta$.

to stronger vertical temperature gradients due to the massive lateral heat transport.

These outstanding heat transport features are now consolidated as laterally-averaged temperature profiles as shown in Fig. 3d, computed using the whole domain, and e, only the left half (dayside). For $\Theta = 0.33$, the 'S'-shaped profile in Fig. 3d is akin to that in RBC, whereas it deviates by compressing the boundary layers while the interior remains mean as $\Theta$ increases. The latter indicates that the imposition of $\Theta$ enhances the vertical heat flux at the surfaces. By looking at only the dayside (Fig. 3e), being aware that the nightside will be the upside-down analog, the mechanism of heat transfer enhancement is unveiled. The overturning circulation transports cold fluids laterally from the nightside, cooling down the dayside as evidenced by the mean temperature profile shifted from 0.5. The heat flux at the bottom thus increases remarkably, whereas the cold bulk fluid thickens the thermal boundary layer at the top, decreasing the heat flux. Ultimately, at high $\Theta$, the heat transfer at the top alters its direction, i.e., heat is removed from the surface.

The convective flows transport heat across the mantle vertically as well as horizontally. The consequential surface heat flux can be characterized using the Nusselt number, Nu, the ratio of convective and diffusive heat transport. We discuss global Nu and local Nu separately as shown in Fig. 3f and g. The former exhibits a distinct dependence on the control parameters. Nu in general increases with Ra and is insensitive to variation of $\Theta$ until $\Theta \leq 1$, leading to the power-law scaling Nu ~ Ra$^{0.11}$. The aforementioned heat transfer enhancement emerges as

the drastic shift of Nu for $\Theta > 1$, showing Nu ~ Ra$^{0.21}$, where the exponent is consistent with the 1/5-scaling in HC[16,24]. Yet, further investigation is needed to derive rigorous scaling laws. The local Nu, decomposed from the global Nu as Nu $= Nu_{top}^{D} + Nu_{bot}^{D} = Nu_{bot}^{N} + Nu_{top}^{N}$, provides an insight into how heat flux is distributed along the horizontal boundaries within the range from $Nu_{top}^{D}$ ($Nu_{bot}^{N}$) to $Nu_{bot}^{D}$ ($Nu_{top}^{N}$). The range widens as $\Theta$ increases, suggesting that the heat flux varies along the surface from the substellar to the antistellar points, and such distributions last permanently.

The value of Nu $= \mathcal{O}(10^1)$ is at the same order of magnitude as that of Earth, $\approx 30$–$60$, which is estimated from the ratio of mantle and plate thicknesses[47]. Although more realistic modeling, including material circulation, is necessary for quantitative discussion, this range of Nu suggests the possibility of the presence of liquid water from the thermodynamic point of view, as lateral heat transport continuously melts the frozen water on the nightside. Furthermore, higher lateral heat flux achieved at large $\Theta$ may result in magma ponds at the surface, potentially forming hydrothermal vents that spark the origin of life[48]. Unlike homogeneously distributed heat flux on the Earth's surface that holds life everywhere, tidally locked exoplanets are expected to localize such moderate heat fluxes at specific regions. Tidal locking appears to force extremely hot and cold environments on surfaces that are intuitively harsh for life. However, such exoplanets may be more tolerant of sustaining life as tidal locking can contribute to maintaining moderate thermal environments locally by distributing heat flux laterally. This

thermodynamic perspective enlightens the future venue for studying the habitability of super-Earth exoplanets.

## Discussion

Building on our experimental results shown in Figs. 2 and 3, we discuss potential scenarios of convective dynamics in mantle layers of tidally locked exoplanets, and their consequential influences on upper surface and deeper core activities. One of the most remarkable features, consistent with earlier numerical studies[14,15], is the perpetual localization of vertical flows in the mantle layers—hot, upward flow persists at the substellar point, and cold, downward flow at the antistellar point. This localization depicts a possible formation of gigantic volcanic mountains at the substellar point on the equator, while yielding little hotspot-type volcanic activities in the mid- and high-latitude regions. Intriguingly, this implication contradicts the conclusions given in previous studies on large super-Earth exoplanets without considering horizontal temperature gradients: The effects of strong adiabatic compression and the associated increase in the adiabatic temperature gradient suppress the upwelling hot plumes emanating from the CMB for larger systems[49,50]. The expected convective dynamics and consequential volcanic activities may differ substantially by the presence of tidal locking. A gigantic volcano will transport substances from the deep mantle to the surface along with its time-dependent volcanic activities discussed in Fig. 2. Such distinctive signatures may manifest as observable quantities like spatial distributions of deposits or atmospheric components[2,4]. In turn, the surface observation may be indirectly conducive to deducing the internal thermal structures.

The spatially anchored thermal structures also sustain the surface environment permanently. Low-latitude regions on the dayside are fully dried up due to their exposure to stellar flux, or ultimately, magma oceans are formed by molten rocks, making them lava worlds with intense stellar flux[51–53]. On the transition from extremely hot dayside to cold nightside, moderate geothermal heat fluxes should exist. The latter is maintained at the mid- and high-latitude regions or in the vicinity of the day-night boundary along a longitude line, potentially enabling them to host liquid water. The presence of liquid water not only depicts potential life but also facilitates active tectonic activities, like the formation of subduction zones, since liquid water softens the lithosphere[54]. Since the water-abundant region is localized, unlike Earth, such tectonic activities are expected to differ vastly from those on Earth.

We also highlight that the impact of the heterogeneous thermal boundaries originating from tidal locking will cascade deeper: the convective dynamics of the liquid core, driven by cooling from the mantle, will be complicated by the horizontal temperature gradients. The experimental results show that $Nu_{bot}^D$ has a larger heterogeneity than the nightside, effectively removing heat from the upper part of the dayside core. But, at the same time, a large downdraft in the nightside center causes cold plume material to be trapped above the nightside CMB, which may affect the cooling of the core[55]. The latter complex convective dynamics of the liquid core influenced by HC will be integrated as an abnormal magnetic field, deviated from the Earth's magnetic dipole. Furthermore, since the rotation and orbital periods are comparable in tidally-locked exoplanets, the Coriolis effect, a key to discuss Earth's dynamo effect[56,57], may become relatively irrelevant. Dynamo effects in such extraordinary conditions remain largely unexplored, triggering further detailed studies. However, the upper-to-lower cascade of tidal locking effect may be irrelevant for massive super-Earths, if radiogenic heating in the core is prominent as proposed recently[58]. The strong internal heating of the core can isolate the core from the persistent cooling due to the downwelling subduction in the mantle, leading to uniform CMB temperature. In this case, the baroclinic torque imposed by $\Delta\theta$—the main driver of HC—presents only at the surface, weakening the lateral (hemispherical) circulations[59].

Although this study focused only on SSSs, it is worth discussing potential scenarios that super-Earth exoplanets commonly experience in their thermal evolution processes. Initial states after giant impacts are expected to form layered convection[60], reducing $\Delta T$ effectively. In such an initial transition, both $\mathcal{A}$ and $\Theta$ are greater than that at the later stage dominated by single-layer convection as discussed in this work. Earlier works found this layered convection to be favored for super-Earths with more than 3–5 Earth masses because high pressures promote endothermic phase transitions, viscosity stratification, and residual chemical layering from magma ocean differentiation[61–63]. The transition to single-layer convection may be accelerated by the lateral heat transport enhanced by the tidal locking. In the course of transitioning, both $\mathcal{A}$ and $\Theta$ will decrease. Eventually, $\Theta$ increases again with time as the system gradually reduces $\Delta T$ for the entire cooling while maintaining $\Delta\theta$, the day-night temperature difference. The synergy effect of RBC and HC reinforcing each other may be diminished by the competition between them for certain scenarios, in particular, for large $\mathcal{A}$ conditions, because of the formation of multiple convective circulations partitioning the system laterally[59]. It is worth addressing that the transition from layered to single-layer convection in the mantle is not only the signature of super-Earth exoplanets, but is also considered for Earth[64]. Moreover, the manifestation of intermittent downward flows, like stagnant slab[65,66], will also impact the lateral length scale of convective flow structures. Addressing how the control parameters, Ra, $\Theta$, and $\mathcal{A}$, evolve and influence the convective dynamics under transient scenarios requires further investigations.

We remark that many of the above aspects must be debated with further investigations since we still lack actual observations of tidally locked super-Earth exoplanets. However, our framework provides a simple but canonical problem setup that can be scaled to delve deeper into convective dynamics of tidally-locked super-Earth exoplanets, hinting at modeling and predicting such faraway worlds' dynamics and consequential environments through a profound physical understanding.

## Methods
### Laboratory experiment

Laboratory analog employs a rectangular fluid tank, $L = 200$ mm, $H = 100$ mm, and $W = H/2 = 50$ mm, enclosed laterally by clear acrylic plates. The top and the bottom surfaces are in contact with copper-made heating/cooling units partitioned by rubber sheets, allowing the step-function-like temperature profiles on the surfaces[33]. Each unit maintains its temperature individually at $T_{bot}^D$, $T_{top}^D$, $T_{bot}^N$, and $T_{top}^N$ by supplying refrigerant from four independent thermostatic baths. This configuration allows stably maintaining $\Delta T$ and $\Delta\theta$ during each experimental run. The details of the fluid tank are described earlier[34]. Room temperature is set $T_0 = 20$ °C, and so is the bulk mean temperature of the system, i.e., $T_{bot}^D = T_0 + \Delta T/2 + \Delta\theta/2$, $T_{top}^D = T_0 - \Delta T/2 + \Delta\theta/2$, $T_{bot}^N = T_0 + \Delta T/2 - \Delta\theta/2$, and $T_{top}^N = T_0 - \Delta T/2 - \Delta\theta/2$, to minimize heat leakage through lateral walls.

To achieve high Pr in the laboratory experiments, we utilized glycerol solutions at three different concentrations, 60-wt.%, 85-wt.%, and 99-wt.%, corresponding to Pr = 88 ($10^2$), 933 ($10^3$), and 9624 ($10^4$) at $T_0$[67], respectively. We seed particles encapsulating thermochromic liquid crystals (TLC) into the fluids for visualization. The $x$-$z$ cross-section at $y = 0.5\,W$ is visualized by a halogen light sheet, showing qualitative temperature distributions by colors[32], as shown in Fig. 1b and Supplementary Movies 1, 2, and 3.

The fluids, initially sitting in the laboratory at room temperature of $T_0$, are well stirred for homogenization and deaerated utilizing a vacuum pump to avoid bubble intrusions. Each experimental run is initiated from a classical RBC case, i.e., only $\Delta T$ is imposed. Once the system develops vigorous RBC after a sufficiently long time, $\Delta\theta$ is imposed at $t = 0$. We continuously record particle motions with a

CMOS color camera before $t = 0$ until the system reaches a statistically steady state with the imposition of $\Delta\theta$.

## Data analysis

Particle images are processed through the in-house image processing scheme for tracking particle motions, which has been utilized elsewhere[33], to quantify velocity fields over successive image frames. Velocity vectors are determined for randomly distributed particles and interpolated on regular grids for later convenience. The consequential spatial resolution is $\approx 0.8$ mm in the $x$ and $z$-directions. The mean temperature fields shown in Fig. 3 are reconstructed by substituting the interpolated mean velocity fields into the steady heat equation as proposed and validated earlier[34,35,45,46]. That is,

$$(\boldsymbol{u} \cdot \nabla)T = \kappa\nabla^2 T,$$

where $\boldsymbol{u}$ is the mean velocity vector obtained from the experiments. The mean temperature distribution $T$ that fulfills the latter equation is found through an iterative process with known boundary conditions. Note that we assume quasi-2D flow fields, i.e., $u_y = 0$ and $\partial u_x/\partial y = \partial u_z/\partial y = \partial T/\partial y = 0$, building on the laboratory observation.

## Data availability

The data to reproduce this paper is provided through https://doi.org/10.6084/m9.figshare.29106215[68]. Any other original data, such as image sequences obtained from the laboratory experiments, are available from the corresponding authors upon request due to the size limitations.

## Code availability

The codes to reproduce this paper are provided through https://doi.org/10.6084/m9.figshare.29106215[68].

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

## Acknowledgements
The authors thank Mr. Toshiyuki Sampo for his technical support in building the experimental setup and Dr. Hugo N. Ulloa for the discussion. The authors also acknowledge two anonymous reviewers for their constructive feedback.

## Author contributions
D.N. led all aspects of this study: conceptualization, experimental design and execution, analysis and interpretation of results, and manuscript drafting. T.M. and T.Y. brought the original idea of the work and participated in data interpretation. T.T. participated in the laboratory experiment and interpretation. Y.T. supervised the work, acquired the funding, and participated in data interpretation. The manuscript was written and reviewed collaboratively by all authors.

## Competing interests
The authors declare no competing interests.
