## [Transparent Peer Review file · Nature Communications]

Convective Dynamics in Mantle of Tidally-Locked Exoplanets

Corresponding Author: Dr Daisuke Noto

Version 0:

Reviewer comments:

Reviewer #1

(Remarks to the Author)

Noto et al. present an elegant analogue experiment designed to explore mantle dynamics in rocky exoplanets subject to strong hemispherical temperature contrasts. Unlike rocky planets in our solar system, which are generally uniformly cooled from the top, tidally locked exoplanets orbiting close to their host stars exhibit a permanent hot day-side and a permanent cold night-side. This thermal asymmetry is expected to drive horizontal convection, superimposed on the vertical convection resulting from the planet's secular cooling. While previous exploratory work have addressed this question, the interplay between horizontal and vertical convection in the exoplanet context remains to be self-consistently investigated and quantified. The authors address this gap through a simple, yet insightful, fluid dynamics laboratory experiment supported by a coherent dimensionless framework.

Their findings can be categorized into two main areas: (a) fluid dynamic insights, and (b) potential applications. On the fluid dynamics side, the authors propose a robust set of dimensionless parameters that capture the combined effects of horizontal ($\Delta\theta$) and vertical (ΔT) thermal forcing. These thermal scales are then used to rederive usual dimensionless numbers relevant to thermal convection, including the Rayleigh, Prandtl, and Reynolds numbers. They further investigate the relationship between the Rayleigh number and the Nusselt number, which is critical for quantifying heat transport efficiency, and would be valuable for the community.

On the application side, the authors suggest that a "moderate geothermal heat flux" located approximately 90 degrees from the antistellar point could be conducive to habitable environments. While this is an intriguing hypothesis, the connection to habitability appears speculative at this stage. For example, it remains unclear why or how water would accumulate at mid-to-high latitudes given the persistent circulation from the day-side to the night-side. Similarly, the implications for surface dynamics and tectonic regimes are not fully addressed.

Overall, this manuscript presents a timely contribution to the study of fluid dynamics in planetary interiors, particularly for tidally locked exoplanets. To my knowledge, the novelty lies in coupling horizontal and vertical thermal forcing in a controlled laboratory setting. However, while the experimental design and fluid dynamic analysis are strong, the broader implications, especially those concerning planetary habitability, require further development and should be treated with appropriate caution. In its current form, the study is most valuable within the exploratory domain of mantle convection in rocky exoplanets or theoretical fluid dynamics.

You will find below more detailed / line by line comments.

1) Vocabulary. Certain word choices could be refined, such as the following:

a. "Perfervid". hot would be preferred.

b. mantle layers. I am not sure that "layers" is required. Why are you using mantle layers although the mantle seems to fully convect and is compositionally homogenous?

2) Line 4. Remove: "Our eternal mystery is".

3) Line 9 to 12. I do not see the logical connection with the sentence before. Also, the phase curve has been revisited since the publication of Demory (2016), and the hot-spot shift was not confirmed. Revisiting the Iconic Spitzer Phase Curve of 55 Cancri e: Hotter Dayside, Cooler Nightside, and Smaller Phase Offset, Mercier et al., 2022., *Astronomical Journal*. DOI 10.3847/1538-3881/ac8f22.

4) Line 22 to 24 need to be rephrased.

- 5) Line 43 to 45. I find the claim of understanding at a “fundamental level” somewhat awkward. I would suggest instead motivating the present research by emphasizing the need to validate numerical simulations through laboratory experiments. Also, other work could be referenced such as Lai, Y., et al., . Ocean circulation on tide-locked lava worlds. (2024a and 2024b) and Boukaré et al., Deep Two-phase, Hemispherical Magma Oceans on Lava Planets (2022). While the above studies investigate a more extreme scenario—where the day-side is hot enough to be molten—the underlying geodynamic principles remain the same as those explored in the present manuscript. By the way, it could be worth to say a few words on rocky planets with extreme day-night temperature contrast (i.e, > 2000 K) from Earth-like case with moderate day-night temperature contrast.
- 6) Line 62-66. “Our findings imply the unexpected potential of tidally-locked exoplanets for supporting life from a thermodynamic point of view [...]” is an overstatement. This is certainly an interesting avenue for further exploration, but in its current form, I believe more is needed to establish a robust connection between local heat flux and habitability.
- 7) Line 68 to 161. The formulation of the problem is clear and easy to follow!
- 8) Line 254. “Increases” may be better than “increments”.
- 9) Line 302-304. While the reference accurately supports the statement, I am not sure that this is universally accepted in the geodynamics community. See for instance, Ricard and Alboussiere, Compressible convection in super-earths, PEPI (2023).
- 10) Figure 3. Figure 3f. Does Ra refer to the hybrid Rayleigh number or the one related to RBC? Would it be possible to derive a Nu-Ra relationship that includes the dependence of theta ? Would you suggest having an exponent that is a function of theta?
- 11) Line 369-386. Again, this is a very interesting outcome that emerges from the study but I do not find this convincing enough to claim for habitability.
- 12) Line 388-395. This is consistent with numerical simulations of Meier et al., 2023.
- 13) Line 477-478. Rephrase. The word “closed” may not be appropriate.

Reviewer #2

(Remarks to the Author)

This study investigates the convective dynamics within the mantle layers of tidally-locked super-Earth exoplanets, which experience extreme day–night temperature contrasts due to permanent stellar illumination on one hemisphere. Using a simplified laboratory analog, the authors model mantle convection under such asymmetric thermal forcing and uncover persistent, system-wide circulation patterns. These convective structures potentially lead to distinctive tectonic and volcanic activity unlike that on Earth. Notably, moderate geothermal fluxes at mid- and high-latitudes suggest the possibility of localized surface liquid water and associated tectonic processes, despite global extremes in stellar heating. The study also explores the implications of mantle dynamics on core convection, magnetic field generation, and long-term thermal evolution. By identifying key dimensionless parameters governing these regimes, the authors provide a scalable framework for understanding the deep interior processes of tidally-locked exoplanets and highlight their potential to sustain habitable environments.

I am not familiar with laboratory setups for investigating mantle convection through optical visualization, so I will assume that the experimental design is well executed and innovative in exploring the hemispheric mantle layer of tidally locked planets. My comments are more from a general perspective on mantle convection in super-Earths.

Recently, Meier et al. (2024) proposed that the mantle dynamics of super-Earth GJ 486b are governed by the strength of the lithosphere and the day-night surface temperature contrast, and Degree-1 convection is a consequence of the strong lithosphere, rather than the temperature contrast between the dayside and nightside. They also conclude that a strong surface temperature contrast between the dayside and nightside can anchor downwellings to one hemisphere. It seems that the authors discuss topics very similar to those in Meier et al. (2024), so a detailed comparison between the current manuscript and Meier et al. (2024) should be provided. Is there any substantial conceptual advance beyond the laboratory realization and the scalable framework?

Line 115. The case of $\Delta T = 3000$ K is discussed. However, for super-Earths with a mass of 10 Earth masses, ΔT can easily exceed 10,000 K. How would this affect the mantle convection scenario?

Line 428-447. A recent paper (Luo et al. 2024) proposes a more realistic core-driven dynamo scenario, where radiogenic heating is primarily concentrated in the core rather than the mantle in massive super-Earths. This could significantly impact the thermal and magnetic evolution of super-Earths. Luo et al. (2024) concludes that mantle convection in super-Earths is primarily driven by heating from the core rather than by a mix of internal heating and cooling from above as in Earth. How does such a scenario affect the convective dynamics in the mantle layers discussed by the authors?

Line 448-476. The authors propose a transition from layered convection to full-depth convection in the mantle. However, several other studies (e.g., Shahnas & Pysklywec, 2021; Stamenković et al. 2012; Spaargaren et al. 2020) suggest that this transition strongly depends on a planet’s size. For super-Earths with more than 3–5 Earth masses, layered mantle convection is expected or favored because high pressures promote endothermic phase transitions, viscosity stratification, and residual chemical layering from magma ocean differentiation. Further discussion on this aspect is needed.

Cited literatures

Meier, T. G., Bower, D. J., Lichtenberg, T., Hammond, M., Tackley, P. J., Pierrehumbert, R. T., et al. (2024). Geodynamics of super-Earth GJ 486b. *Journal of Geophysical Research: Planets*, 129, e2024JE008491.

H. Luo, J. O'Rourke, J. Deng, Radiogenic heating sustains long-lived volcanism and magnetic dynamos in super-Earths. *Sci. Adv.* 10, (2024).

Shahnas, M. H., & Pysklywec, R. N. (2021). Focused penetrative plumes: A possible consequence of the dissociation transition of post-perovskite at ~ 0.9 TPa in massive rocky super-Earths. *Geochemistry, Geophysics, Geosystems*, 22, e2021GC009910

Vlada Stamenković, Lena Noack, Doris Breuer, and Tilman Spohn, The Influence of Pressure-dependent Viscosity on the Thermal Evolution of Super-Earths. *The Astrophysical Journal*, 41, 22 pp. (2012).

Rob J. Spaargaren, Maxim D. Ballmer, Dan J. Bower, Caroline Dorn and Paul J. Tackley, The influence of bulk composition on the long-term interior-atmosphere evolution of terrestrial exoplanets. *A&A*, 643 (2020) A44

Answers to Reviewer #1

Overall Comment: Noto et al. present an elegant analogue experiment designed to explore mantle dynamics in rocky exoplanets subject to strong hemispherical temperature contrasts. Unlike rocky planets in our solar system, which are generally uniformly cooled from the top, tidally locked exoplanets orbiting close to their host stars exhibit a permanent hot day-side and a permanent cold night-side. This thermal asymmetry is expected to drive horizontal convection, superimposed on the vertical convection resulting from the planet's secular cooling. While previous exploratory work have addressed this question, the interplay between horizontal and vertical convection in the exoplanet context remains to be self-consistently investigated and quantified. The authors address this gap through a simple, yet insightful, fluid dynamics laboratory experiment supported by a coherent dimensionless framework.

Their findings can be categorized into two main areas: (a) fluid dynamic insights, and (b) potential applications. On the fluid dynamics side, the authors propose a robust set of dimensionless parameters that capture the combined effects of horizontal ($\Delta\theta$) and vertical (ΔT) thermal forcing. These thermal scales are then used to rederive usual dimensionless numbers relevant to thermal convection, including the Rayleigh, Prandtl, and Reynolds numbers. They further investigate the relationship between the Rayleigh number and the Nusselt number, which is critical for quantifying heat transport efficiency, and would be valuable for the community.

On the application side, the authors suggest that a “moderate geothermal heat flux” located approximately 90 degrees from the antistellar point could be conducive to habitable environments. While this is an intriguing hypothesis, the connection to habitability appears speculative at this stage. For example, it remains unclear why or how water would accumulate at mid-to-high latitudes given the persistent circulation from the day-side to the night-side. Similarly, the implications for surface dynamics and tectonic regimes are not fully addressed.

Overall, this manuscript presents a timely contribution to the study of fluid dynamics in planetary interiors, particularly for tidally locked exoplanets. To my knowledge, the novelty lies in coupling horizontal and vertical thermal forcing in a controlled laboratory setting. However, while the experimental design and fluid dynamic analysis are strong, the broader implications, especially those concerning planetary habitability, require further development and should be treated with appropriate caution. In its current form, the study is most valuable within the exploratory domain of mantle convection in rocky exoplanets or theoretical fluid dynamics.

You will find below more detailed / line by line comments.

We would like to express our gratitude for the reviewer's critical but constructive feedback. Also, we thank the reviewer for making comments while listing relevant literature that we have overlooked. The comments are very fair and allow us to deepen our understanding of existing

literature, helping us better rephrase our text while avoiding overstating the applications. In particular, we have provided too bold extrapolations regarding habitability, which should have accounted for other aspects that our framework does not integrate. We have carefully revisited the literature and the original manuscript and made appropriate corrections.

We will now proceed to address the reviewer's comments individually.

Comment 1: 1) Vocabulary. Certain word choices could be refined, such as the following:
a. "Perfervid". hot would be preferred. b. mantle layers. I am not sure that "layers" is required. Why are you using mantle layers although the mantle seems to fully convect and is compositionally homogenous?

Thank you for pointing out the vocabulary, which is not appropriate. We have removed 'perfervid' from the text. We also agree with the reviewer's suggestion on 'layers'. We removed them to avoid confusion. Because of this, we changed the title to "Convective Dynamics in Mantle of Tidally-Locked Exoplanets".

Comment 2: 2) Line 4. Remove: "Our eternal mystery is".

Thank you for the suggestions. We changed the corresponding expression to 'A simple yet profoundly intriguing question is'.

Comment 3: 3) Line 9 to 12. I do not see the logical connection with the sentence before. Also, the phase curve has been revisited since the publication of Demory (2016), and the hot-spot shift was not confirmed. Revisiting the Iconic Spitzer Phase Curve of 55 Cancri e: Hotter Dayside, Cooler Nightside, and Smaller Phase Offset, Mercier et al., 2022., *Astronomical Journal*. DOI 10.3847/1538-3881/ac8f22.

Thank you for pointing out the sentences that are not clearly linked to each other. Also, thank you for providing us with the important paper to be cited. What we would like to state in this paragraph is the large variability of the surface conditions on super-Earth exoplanets, because various scenarios have been observed even for planets having the most detailed observations like 55 Cancri e and LHS 3844 b, enumerated in the main text. This paragraph later helps us justify why the problem can be formulated by considering only the consequential temperature difference ΔT and $\Delta\theta$, instead of combinations of other observable quantities, such as stellar flux and atmospheric conditions.

We substantially rewrote the introduction and some related sentences later to better introduce the need for the canonical framework of the problem in a way that we can scale to any potential scenarios supported by the present-day observations.

Comment 4: 4) Line 22 to 24 need to be rephrased.

Thank you for the suggestions. The corresponding text is largely modified to align with the new context of the introduction.

Comment 5: 5) Line 43 to 45. I find the claim of understanding at a “fundamental level” somewhat awkward. I would suggest instead motivating the present research by emphasizing the need to validate numerical simulations through laboratory experiments. Also, other work could be referenced such as Lai, Y., et al., . Ocean circulation on tide-locked lava worlds. (2024a and 2024b) and Boukaré et al., Deep Two-phase, Hemispherical Magma Oceans on Lava Planets (2022). While the above studies investigate a more extreme scenario—where the day-side is hot enough to be molten—the underlying geodynamic principles remain the same as those explored in the present manuscript. By the way, it could be worth to say a few words on rocky planets with extreme day-night temperature contrast (i.e, > 2000 K) from Earth-like case with moderate day-night temperature contrast.

Thank you for the comment. We agree with the reviewer, the expression ‘fundamental level’ is not clear enough. We also agree with the reviewer’s comment about the need to validate numerical simulations. However, we consider that one of the key contributions of our work is providing a dimensionless framework of mantle convection in tidally-locked exoplanets. We acknowledge that some numerical works have already showcased convective dynamics earlier [1–3]; these works focused on specific exoplanets. By offering a dimensionless framework that can be extended to any similar systems, we can generalize and unify the physical interpretations established for such specific systems. Also, our experimental results obtained in the laboratory experiments, exhibiting similar dynamics to those in existing works, are evidence that the framework is scalable through the dimensionless parameters we proposed. Accordingly, we consider the value of the present work to be not only the validation of numerical simulations, but also the generalization of the problem.

Thank you for providing us with more references [4–6]. We carefully checked them and now they are implemented in the discussion section of the revised manuscript. These extreme stellar flux conditions, leading to form lava worlds, are not explorable in laboratory experiments and are interesting to discuss to further extend the work.

Extreme scenarios are interesting aspects to further discuss. We now mention this extreme condition in the problem formulation to offer a physical sense of possible ranges of ΔT and $\Delta\theta$, instead of introducing specific examples. However, these specific values are not relevant later because all discussion is based on dimensionless parameters.

Comment 6: 6) Line 62-66. “Our findings imply the unexpected potential of tidally-locked exoplanets for supporting life from a thermodynamic point of view [. . .]” is an overstatement.

This is certainly an interesting avenue for further exploration, but in its current form, I believe more is needed to establish a robust connection between local heat flux and habitability.

Thank you for the comment. We fully agree with this comment; more needs to be established from the perspective that we did not integrate into the framework to make a firm conclusion on habitability. We have rephrased the corresponding text, resulting in a tone down for the potential habitability, but a more precise description that delineates a clear venue for future investigation. This modification helps us improve the accuracy of the interpretation while expanding its implications.

Comment 7: 7) Line 68 to 161. The formulation of the problem is clear and easy to follow!

Thank you for recognizing the value of this part. We are happy to see this comment.

Comment 8: 8) Line 254. “Increases” may be better than “increments”.

Thank you for the suggestions. We changed it.

Comment 9: 9) Line 302-304. While the reference accurately supports the statement, I am not sure that this is universally accepted in the geodynamics community. See for instance, Ricard and Alboussiere, Compressible convection in super-earths, PEPI (2023).

Thank you for the comment and for suggesting the key paper [7]. It is interesting to further discuss the potential convective dynamics in super-Earths, building on more complex models of rheology or equations of state. We recognize that our description was relatively biased by one specific paper from one of the authors, and more should have been considered to improve universality. We now explicitly state different conclusions derived from different modeling approaches, but in the absence of horizontal convection. However, the conclusion regarding the reinforcement of volcanic/tectonic activities by the addition of horizontal convection remains the same here, and we believe that it does not contradict any existing literature.

Comment 10: 10) Figure 3. Figure 3f. Does Ra refer to the hybrid Rayleigh number or the one related to RBC? Would it be possible to derive a Nu-Ra relationship that includes the dependence of theta? Would you suggest having an exponent that is a function of theta?

Thank you for the comment. Figures. 3f and g both refer to the hybrid Rayleigh number. In our manuscript, we consistently use Ra for the hybrid Rayleigh number, and Ra_z for the standard Rayleigh number that is usually used for Rayleigh–Bénard convection (RBC).

Unfortunately, we have not established the scaling for the Nu-Ra relationship yet. This is certainly our future work. Since the Nu-Ra_z and Nu-Ra_x scaling relationship have been extensively studied [8, 9], we expect to construct similar relationships using Ra_z and Ra_x. Since their forms have not been established yet, even in the other numerical work [10], we cannot suggest any formulation. What we can anticipate is that for $\Theta \ll 1$, the system becomes Rayleigh–Bénard-like, characterized by $\text{Nu} \sim \text{Ra}_z^{1/3}$, and for $\Theta \gg 1$ the system becomes horizontal convection-like, characterized by $\text{Nu} \sim \text{Ra}_x^{1/5}$. In fact, the latter signature is already captured in Fig. 3f. However, in the transition regime, where we investigated $\Theta \sim 1$, we anticipate that the heat transport scaling exhibits intertwined effects of both Rayleigh numbers because of the competition/cooperation of Rayleigh–Bénard and horizontal convection. We briefly remark importance of this part in *Supplementary Information* to trigger future investigations.

Comment 11: 11) Line 369-386. Again, this is a very interesting outcome that emerges from the study but I do not find this convincing enough to claim for habitability.

Thank you for pointing this out. Although more aspects must be considered in future investigations to discuss further the habitability, our work offers a thermodynamic point of view on it. We have now changed the tone of the description. Instead of highlighting the potential of habitability, we put emphasis on the presence of moderate thermal environments, which render the potential presence of liquid water, which may sustain life. This expression better frames the value of our findings.

Comment 12: 12) Line 388-395. This is consistent with numerical simulations of Meier et al., 2023.

Thank you for the comment. We have overlooked this paper [11], and now it is incorporated in the revised manuscript to support our statement.

Comment 13: 13) Line 477-478. Rephrase. The word “closed” may not be appropriate.

Thank you for the suggestions. We rephrased the corresponding sentence.

Answers to Reviewer #2

Overall Comment: This study investigates the convective dynamics within the mantle layers of tidally-locked super-Earth exoplanets, which experience extreme day–night temperature contrasts due to permanent stellar illumination on one hemisphere. Using a simplified laboratory analog, the authors model mantle convection under such asymmetric thermal forcing and uncover persistent, system-wide circulation patterns. These convective structures potentially lead to distinctive tectonic and volcanic activity unlike that on Earth. Notably, moderate geothermal fluxes at mid- and high-latitudes suggest the possibility of localized surface liquid water and associated tectonic processes, despite global extremes in stellar heating. The study also explores the implications of mantle dynamics on core convection, magnetic field generation, and long-term thermal evolution. By identifying key dimensionless parameters governing these regimes, the authors provide a scalable framework for understanding the deep interior processes of tidally-locked exoplanets and highlight their potential to sustain habitable environments.

I am not familiar with laboratory setups for investigating mantle convection through optical visualization, so I will assume that the experimental design is well executed and innovative in exploring the hemispheric mantle layer of tidally locked planets. My comments are more from a general perspective on mantle convection in super-Earths.

We are grateful for the reviewer’s positive and constructive feedback. We have been working on this kind of laboratory experiment for decades, accumulating methods and practical knowledge to perform reliable laboratory experiments. Some of the related works are cited in the main text, and they have been approved and published in journals mostly from the fluid mechanics community and the experimental community of optical visualization. The experiments were performed with scrupulous care to the best of our knowledge. Moreover, although the data is not shown in order to self-contain this manuscript as an experimental work, we have conducted three-dimensional direct numerical simulations to validate the experimental results (to be reported in the future), showing great agreement with the experimental results.

We will now proceed to address the reviewer’s comments individually.

Comment 1: Recently, Meier et al. (2024) proposed that the mantle dynamics of super-Earth GJ 486b are governed by the strength of the lithosphere and the day-night surface temperature contrast, and Degree-1 convection is a consequence of the strong lithosphere, rather than the temperature contrast between the dayside and nightside. They also conclude that a strong surface temperature contrast between the dayside and nightside can anchor downwellings to one hemisphere. It seems that the authors discuss topics very similar to those in Meier et al. (2024), so a detailed comparison between the current manuscript and Meier et al. (2024) should be provided. Is there any substantial conceptual advance beyond

the laboratory realization and the scalable framework?

We appreciate this comment, which indicates a highly relevant paper to our work, yet was overlooked in the original manuscript. The combinations of the strength of the lithosphere and the lateral temperature contrasts are essential, although it is, unfortunately, (almost) impossible to implement in a laboratory setup. Technically speaking, this is simply because we cannot find a reasonable yield stress fluid whose rheological effect becomes effective with the strength of thermal forcing under laboratory environments. Also, it may be possible to see our work as the case with a strong lithosphere since the surface boundary condition is no-slip (literally a stagnant lid), and the test fluid represents the convecting asthenosphere. In this way, our framework does not contradict with Meier *et al.* (2024) [3]. Although we have overlooked this paper, we are happy to see this work, whose conclusion aligns well with ours, in light of the formation of hemispheric circulations due to strong day-night temperature contrasts.

Before knowing this paper, we considered one of the biggest conceptual advancements to be the direct consideration of the temperature conditions rather than the stellar flux conditions like models proposed in Meier *et al.* (2021) [2]. This is because — as Meier *et al.* (2024) [3] also states similar — the surface atmospheric environments are highly system-dependent, and the consequential surface temperature may be better parametrized to expand the applicability of the framework for mantle convection to various potential scenarios. This simplifies the problem setting by eliminating the estimation of thermal boundary conditions that are actually imposed on the mantle, as a consequence of the combined effects of stellar fluxes and atmospheric conditions. Although this paper [3] already exists, we still see a noticeable advancement from our framework, as we generalized the problem further by defining all parameters dimensionless. This is in particular significant when extending the physical findings obtained earlier to unknown systems, as interpretations are scalable through dimensionless parameters. The experimental realization itself is novel, yet its most important role is offering proof of the scalable framework that allows us to derive the same conclusions obtained in specific systems like those in Meier *et al.* (2024) [3].

Comment 2: Line 115. The case of $\Delta T = 3000$ K is discussed. However, for super-Earths with a mass of 10 Earth masses, ΔT can easily exceed 10,000 K. How would this affect the mantle convection scenario?

Thank you for this comment. We agree with the reviewer’s comment, the vertical temperature difference differs depending on the size of the system. The plausible ranges of ΔT and $\Delta \theta$ are somewhat constrained, but their combinations are extremely system-dependent and are largely uncertain. Also, in the dimensionless framework, discussing specific values of temperature difference is not the most efficient, because the governing parameters are Ra and Θ , rather than ΔT and $\Delta \theta$ themselves. In this context, we expect for instance a planet with $\Delta T = 1000$ K and that with 10000 K can exhibit the same dynamics if the combination of Ra and Θ is identical.

However, foreseeing applications to massive super-Earths, we may need to tune the problem

formulation so that it can parameterize additional complexity which can arise in specific systems. For instance, we assume that super-Earths with a mass of 10 Earth masses experience strong adiabatic compression at deep mantles due to extraordinary pressure at a few TPa [12, 13]. Because of this, the increase of the potential temperature, which actually contributes to buoyancy driving the convective dynamics, may not be as high as that of the temperature. That is, the effective increase of buoyancy is not proportional to the increase of temperature for the cases of gigantic super-Earths. Accordingly, the resultant effect in thermal forcing in vertical direction is not significantly large even if we consider $\Delta T = 10000$ K, instead of $\Delta T = 3000$ K — therefore, the range of the effective vertical Rayleigh number Ra_z will remain the same order of magnitude.

Regarding Θ , $\Delta T = 10000$ K suggests the lower limit, $\Theta = \Delta\theta/\Delta T = 0.1$, for $\Delta\theta = 1000$ K. Yet, this lower limit may increase effectively when we utilize the potential temperature difference for ΔT . Accordingly, we anticipate that the effect of the lateral gradient may become slightly more important under this scenario, but not substantially.

Although we do not expand this discussion specifically in the main text, we added the potential range of ΔT in the problem formulation so that readers can acquire a physical sense of the possible ranges of ΔT and $\Delta\theta$.

Comment 3: Line 428-447. A recent paper (Luo et al. 2024) proposes a more realistic core-driven dynamo scenario, where radiogenic heating is primarily concentrated in the core rather than the mantle in massive super-Earths. This could significantly impact the thermal and magnetic evolution of super-Earths. Luo et al. (2024) concludes that mantle convection in super-Earths is primarily driven by heating from the core rather than by a mix of internal heating and cooling from above as in Earth. How does such a scenario affect the convective dynamics in the mantle layers discussed by the authors?

Thank you for the comment on the effect of internal heating in the core, which we have not discussed in the original manuscript. It is an interesting aspect that we should discuss in the main text. We anticipate that two effects of the radiogenic heating inside the core will appear.

One is simply the increase of the core temperature as concluded in Luo *et al.* (2024) [14], which will be integrated as the increase of ΔT in the present study. Yet, since our discussion is already based on the consequential temperature difference, we can interpret that the internal heating in the core is already accounted for, even though it was not explicitly mentioned in the main text.

Another effect, which is more relevant to consider, is the homogenization of the core temperature and that at the core-mantle boundary (CMB). If the internal heating is strong enough, the convective dynamics in the core can be isolated from the persistent surface cooling, anchoring downwelling flows in the mantle, on the nightside. The core temperature will be homogenized by the vigorous convective mixing, resulting in little thermal nonuniformity at the CMB. In this case, it may not be appropriate to consider the same Θ (or $\Delta\theta$) at the surface and the CMB. A similar setup was presented in Coustou *et al.* (2021) [10], where they numerically considered a constant heat flux at the bottom and a linear horizontal temperature gradient at the top in a wide domain (the

length-to-height aspect ratio 8, corresponding to the mantle thickness relative to the planet radius $\phi \approx 0.328$), although they considered water as the test fluid, $Pr = O(10)$. The convective dynamics at steady states is expected to be similar when the horizontal temperature gradient is strong enough to drive the overturning circulation — the system resembles horizontal convection. By contrast, when the horizontal temperature gradient is weak relative to the vertical one, the system can sustain local convective cells, akin to some cases reported in Meier *et al.* (2021, 2024) [2, 3]. These competitions between the vertical Rayleigh–Bénard-like convection and the horizontal convection will become more relevant for the uniform CMB temperature. This is because our setup, both the surface and the CMB are nonuniform, promoting the system-scale overturning circulation by the baroclinic torque imposed on the boundaries, whereas only the surface does for the uniform CMB temperature conditions.

Our modifications for this comment are: First, acknowledging that the radiogenic internal heating in the core can increase the core temperature, we clarify that our setting considers ΔT and $\Delta\theta$ to be the consequential temperature differences accounting for all possible effects. Second, we deepen the discussion on the potential effects expected to happen in the cases with significant internal core heating, and how it can be incorporated in the present framework in future studies.

Comment 4: Line 448-476. The authors propose a transition from layered convection to full-depth convection in the mantle. However, several other studies (e.g., Shahnas and Pysklywec, 2021; Stamenković *et al.* 2012; Spaargaren *et al.* 2020) suggest that this transition strongly depends on a planet’s size. For super-Earths with more than 3–5 Earth masses, layered mantle convection is expected or favored because high pressures promote endothermic phase transitions, viscosity stratification, and residual chemical layering from magma ocean differentiation. Further discussion on this aspect is needed.

Thank you for the suggestions. In the original manuscript, we did not consider specific conditions of planet sizes; however, it is important to discuss these aspects more precisely, building on the literature. We carefully checked the papers that the reviewer suggested [15–17] and incorporated them into the discussion.

Cited references by Reviewer 2:

- Meier, T. G., Bower, D. J., Lichtenberg, T., Hammond, M., Tackley, P. J., Pierrehumbert, R. T., *et al.* (2024). Geodynamics of super-Earth GJ 486b. *Journal of Geophysical Research: Planets*, 129, e2024JE008491
- H. Luo, J. O’Rourke, J. Deng, Radiogenic heating sustains long-lived volcanism and magnetic dynamos in super-Earths. *Sci. Adv.* 10, (2024)
- Shahnas, M. H., and Pysklywec, R. N. (2021). Focused penetrative plumes: A possible

consequence of the dissociation transition of post-perovskite at ~ 0.9 TPa in massive rocky super-Earths. *Geochemistry, Geophysics, Geosystems*, 22, e2021GC009910

- Vlada Stamenković, Lena Noack, Doris Breuer, and Tilman Spohn, The Influence of Pressure-dependent Viscosity on the Thermal Evolution of Super-Earths. *The Astrophysical Journal*, 41, 22 pp. (2012)
- Rob J. Spaargaren, Maxim D. Ballmer, Dan J. Bower, Caroline Dorn and Paul J. Tackley, The influence of bulk composition on the long-term interior-atmosphere evolution of terrestrial exoplanets. *A&A*, 643 (2020) A44

Thank you for explicitly listing the papers you cited. This helps us easily find relevant papers that we have overlooked, and they are implemented in the revised manuscript.

References

- [1] S. Gelman, L. Elkins-Tanton, and S Seager, “Effects of stellar flux on tidally locked terrestrial planets: Degree-1 mantle convection and local magma ponds,” *Astrophys. J.*, vol. 735, no. 2, p. 72, 2011. DOI: [10.1088/0004-637X/735/2/72](https://doi.org/10.1088/0004-637X/735/2/72).
- [2] T. G. Meier, D. J. Bower, T. Lichtenberg, P. J. Tackley, and B.-O. Demory, “Hemispheric tectonics on super-Earth LHS 3844b,” *Astrophys. J. Lett.*, vol. 908, no. 2, p. L48, 2021. DOI: [10.3847/2041-8213/abe400](https://doi.org/10.3847/2041-8213/abe400).
- [3] T. G. Meier, D. J. Bower, T. Lichtenberg, M. Hammond, P. J. Tackley, R. T. Pierrehumbert, J. A. Caballero, S.-M. Tsai, M. Weiner Mansfield, N. Tosi, and P. Baumeister, “Geodynamics of super-Earth GJ 486b,” *J. Geophys. Res. Planets*, vol. 129, no. 10, e2024JE008491, 2024. DOI: [10.1029/2024JE008491](https://doi.org/10.1029/2024JE008491).
- [4] Y. Lai, J. Yang, and W. Kang, “Ocean Circulation on Tide-locked Lava Worlds. I. An Idealized 2D Numerical Model,” *Planet. Sci. J.*, vol. 5, no. 9, p. 204, 2024. DOI: [10.3847/PSJ/ad7111](https://doi.org/10.3847/PSJ/ad7111).
- [5] Y. Lai, W. Kang, and J. Yang, “Ocean Circulation on Tide-locked Lava Worlds. II. Scalings,” *Planet. Sci. J.*, vol. 5, no. 9, p. 205, 2024. DOI: [10.3847/PSJ/ad70b4](https://doi.org/10.3847/PSJ/ad70b4).
- [6] C.-É. Boukaré, N. B. Cowan, and J. Badro, “Deep two-phase, hemispherical magma oceans on lava planets,” *Astrophys. J.*, vol. 936, no. 2, p. 148, 2022. DOI: [10.3847/1538-4357/ac8792](https://doi.org/10.3847/1538-4357/ac8792).
- [7] Y. Ricard and T. Alboussière, “Compressible convection in super-earths,” *Phys. Earth Planet. Inter.*, vol. 341, p. 107 062, 2023. DOI: [10.1016/j.pepi.2023.107062](https://doi.org/10.1016/j.pepi.2023.107062).
- [8] G. Ahlers, S. Grossmann, and D. Lohse, “Heat transfer and large scale dynamics in turbulent Rayleigh–Bénard convection,” *Rev. Mod. Phys.*, vol. 81, no. 2, p. 503, 2009. DOI: [10.1103/RevModPhys.81.503](https://doi.org/10.1103/RevModPhys.81.503).
- [9] G. O. Hughes and R. W. Griffiths, “Horizontal convection,” *Annu. Rev. Fluid Mech.*, vol. 40, pp. 185–208, 2008. DOI: [10.1146/annurev.fluid.40.111406.102148](https://doi.org/10.1146/annurev.fluid.40.111406.102148).
- [10] L.-A. Coustou, J. Nandaha, and B. Favier, “Competition between Rayleigh–Bénard and horizontal convection,” *J. Fluid Mech.*, vol. 947, A13, 2022. DOI: [10.1017/jfm.2022.613](https://doi.org/10.1017/jfm.2022.613).
- [11] T. G. Meier, D. J. Bower, T. Lichtenberg, M. Hammond, and P. J. Tackley, “Interior dynamics of super-Earth 55 Cancri e,” *Astron. Astrophys.*, vol. 678, A29, 2023. DOI: [10.1051/0004-6361/202346950](https://doi.org/10.1051/0004-6361/202346950).
- [12] D. Valencia, R. J. O’Connell, and D. Sasselov, “Internal structure of massive terrestrial planets,” *Icarus*, vol. 181, no. 2, pp. 545–554, 2006. DOI: [10.1016/j.icarus.2005.11.021](https://doi.org/10.1016/j.icarus.2005.11.021).
- [13] M. Shahnas and R. Pysklywec, “Penetrative Superplumes in the Mantle of Large Super-Earth Planets: A Possible Mechanism for Active Tectonics in the Massive Super-Earths,” *Geochem. Geophys. Geosyst.*, vol. 24, no. 2, e2022GC010678, 2023. DOI: [10.1029/2022GC010678](https://doi.org/10.1029/2022GC010678).

- [14] H. Luo, J. G. O'Rourke, and J. Deng, "Radiogenic heating sustains long-lived volcanism and magnetic dynamos in super-Earths," *Sci. Adv.*, vol. 10, no. 37, eado7603, 2024. doi: [10.1126/sciadv.ado7603](https://doi.org/10.1126/sciadv.ado7603).
- [15] V. Stamenković, L. Noack, D. Breuer, and T. Spohn, "The influence of pressure-dependent viscosity on the thermal evolution of super-Earths," *Astrophys. J.*, vol. 748, no. 1, p. 41, 2012. doi: [10.1088/0004-637X/748/1/41](https://doi.org/10.1088/0004-637X/748/1/41).
- [16] R. J. Spaargaren, M. D. Ballmer, D. J. Bower, C. Dorn, and P. J. Tackley, "The influence of bulk composition on the long-term interior-atmosphere evolution of terrestrial exoplanets," *Astron. Astrophys.*, vol. 643, A44, 2020. doi: [10.1051/0004-6361/202037632](https://doi.org/10.1051/0004-6361/202037632).
- [17] M. Shahnas and R. Pysklywec, "Focused penetrative plumes: A possible consequence of the dissociation transition of post-perovskite at ~0.9 TPA in massive rocky super-earths," *Geochem. Geophys. Geosyst.*, vol. 22, no. 8, e2021GC009910, 2021. doi: [10.1029/2021GC009910](https://doi.org/10.1029/2021GC009910).